# RoCourseNet: Robust Training of a Prediction Aware Recourse Model

## Abstract

Counterfactual (CF) explanations for machine learning (ML) models are preferred by end-users, as they explain the predictions of ML models by providing a recourse (or contrastive) case to individuals who are adversely impacted by predicted outcomes. Existing CF explanation methods generate recourses under the assumption that the underlying target ML model remains stationary over time. However, due to commonly occurring distributional shifts in training data, ML models constantly get updated in practice, which might render previously generated recourses invalid and diminish end-users trust in our algorithmic framework. To address this problem, we propose RoCourseNet, a training framework that jointly optimizes for predictions and recourses that are robust to future data shifts. We have three main contributions: (i) We propose a novel *virtual data shift (VDS)* algorithm to find worst-case shifted ML models by explicitly considering the worst-case data shift in the training dataset. (ii) We leverage adversarial training to solve a novel tri-level optimization problem inside RoCourseNet, which simultaneously generates predictions and corresponding robust recourses. (iii) Finally, we evaluate RoCourseNet's performance on three real-world datasets and show that RoCourseNet outperforms state-of-the-art baselines by ∼10% in generating robust CF explanations.

## 1 Introduction

To explain the prediction made by a Machine Learning (ML) model on data point $x$, counterfactual (CF) explanation methods find a new *counterfactual* example $x^{\text{cf}}$, which is similar to $x$ but gets a different/opposite prediction from the ML model. CF explanations (Wachter et al., 2017; Karimi et al., 2020; Verma et al., 2020) are often preferred by end-users as they provide actionable recourse[1] to individuals who are negatively impacted by algorithm-mediated decisions. For example, CF explanation techniques can provide recourse for impoverished loan applicants whose loans have been denied by a bank's ML algorithm.

Most CF explanation techniques assume that the underlying ML model is stationary and does not change over time (Barocas et al., 2020). However, in practice, ML models are often updated regularly when new data is available to improve predictive accuracy on the new shifted data distribution. This shifted ML model might render previously recommended recourses ineffective (Rawal et al., 2020), and in turn, diminish end users' trust towards our system. For example, when providing a recourse to a loan applicant who was denied a loan by the bank's ML algorithm, it is critical to approve re-applications that fully follow recourse recommendations, even if the bank updates their ML model in the meantime. This necessitates the development of robust algorithms that can generate recourses which remain effective (or valid) for an end-user in the face of ML models being frequently updated. Figure 1 illustrates this challenge of generating robust recourses.

**Limitations of Prior Work.** To our knowledge, only two studies (Upadhyay et al., 2021; Nguyen et al., 2022) propose algorithmic methods to generate robust recourses. Unfortunately, both these studies suffer from two major limitations. First, both methods are based on strong modeling assumptions which degrades their effectiveness at finding robust recourses (as we show in Section 4). For example, Upadhyay et al. (2021) assume that the ML model's decision boundary can be

---

[1] Note that counterfactual explanation (Wachter et al., 2017) and algorithmic recourse (Ustun et al., 2019) are closely related (Verma et al., 2020; Stepin et al., 2021). Hence, we use these terms interchangeably.

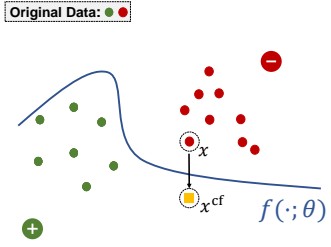 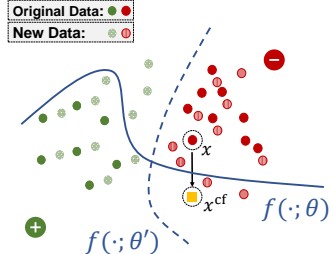 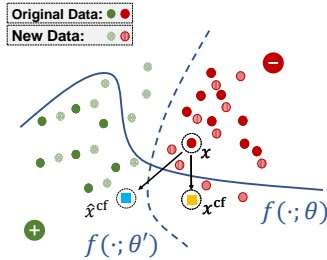

(a) The decision boundary of model $f(\cdot, \theta)$ trained on the original data, and a recourse $x^{\text{cf}}$ of the example $x$ generated CF explanation methods.

(b) Updated decision boundary of retrained new model $f(\cdot, \theta')$ with newly available data (or a shifted data distribution).

(c) Under the shifts in data distribution and model, the recourse $x^{\text{cf}}$ becomes invalid, but $\hat{x}^{\text{cf}}$ is valid. We call $\hat{x}^{\text{cf}}$ as a robust recourse.

Figure 1: **Illustration of the robust recourse generation.** (a) Given an input data point $x$, CF explanation methods generate a new recourse $x^{\text{cf}}$ which lies on the opposite side of decision boundary $f(.; \theta)$. (b) As *new data* is made available, the ML model's decision boundary is updated as $f(.; \theta')$. This shifted decision boundary $f(.; \theta')$ invalidates the chosen recourse $x^{\text{cf}}$ (as $x$ and $x^{\text{cf}}$ lie on the same side of the shifted model $f(.; \theta')$). (c) However, robust CF explanation methods generate a *robust* recourse $\hat{x}^{\text{cf}}$ for input $x$ by anticipating the future shifted model $f(.; \theta')$.

locally approximated via a linear function, and adopt LIME (Ribeiro et al., 2016) to find this linear approximation. However, recent works show that the local approximation generated from LIME is unfaithful (Laugel et al., 2018; Rudin, 2019) and inconsistent (Slack et al., 2020; Alvarez-Melis & Jaakkola, 2018). Similarly, Nguyen et al. (2022) assumes that the underlying data distribution can be approximated using kernel density estimators (Bickel et al., 2009). However, kernel density estimators suffers from the *curse of dimensionality* (Bellman, 1961), i.e., they perform exponentially worse with increasing dimensionality of data (Crabbe, 2013; Nagler & Czado, 2016), which limits its usability in estimating the data distributions of real-world high-dimensional data.

Second, these two techniques are post-hoc methods designed for use with proprietary black-box ML models whose training data and model weights are not available. However, with the advent of data regulations that enshrine the "*Right to Explanation*" (e.g., EU-GDPR (Wachter et al., 2017)), service providers are required by law to communicate both the decision outcome (i.e., the ML model's prediction) and its actionable implications (i.e., a recourse for this prediction) to an end-user. In these scenarios, the post-hoc assumption is overly limiting, as service providers can build recourse models that leverage the knowledge of their ML model to generate higher-quality recourses. In fact, prior work (Guo et al., 2021) has shown that post-hoc CF explanation approaches are unable to balance the cost-invalidity trade-off (Rawal et al., 2020), which is an important consideration in generating recourses. To date, very little prior work departs from the post-hoc paradigm; Guo et al. (2021) propose one such approach, unfortunately, it does not consider the robustness of generated recourses.

**Contributions.** We propose *Ro*bust Re*Course* Neural *Net*work (or RoCourseNet), a novel algorithmic framework for generating recourses which: (i) departs from the prevalent post-hoc paradigm of generating recourses; while (ii) explicitly optimizing the robustness of the generated recourses. RoCourseNet makes the following three novel contributions:

- (Formulation-wise) We formulate the robust recourse generation problem as a tri-level (min-max-min) optimization problem, which consists of two sub-problems: (i) a bi-level (max-min) problem which simulates a worst-case attacker to find an adversarially shifted model by explicitly simulating the *worst-case data shift* in the training dataset; and (ii) an outer minimization problem which simulates an ML model designer who wants to generate robust recourses against this worst-case bi-level attacker. Unlike prior approaches, our bi-level attacker formulation explicitly connects shifts in the underlying data distribution to corresponding shifts in the ML model parameters.

- (Methodology-wise) We propose *RoCourseNet* for solving our tri-level optimization problem for generating robust recourses. RoCourseNet relies on two key ideas: (i) we propose a novel *Virtual Data Shift (VDS)* algorithm to optimize for the inner bi-level (max-min) attacker problem, which results in an adversarially shifted model; and (ii) inspired by Guo et al. (2021), RoCourseNet leverages a block-wise coordinate descent training procedure to optimize the robustness of generated

recourses against these adversarially shifted models. Unlike prior methods (Upadhyay et al., 2021), we require no intermediate steps in approximating the underlying model or data distribution.

- (Experiment-wise) We conduct rigorous experiments on three real-world datasets to evaluate the robustness of several popular recourse generation methods under data shifts. Our results show that RoCourseNet generates highly robust CF explanations against data shifts, as it consistently achieves >96% robust validity, outperforming state-of-the-art baselines by ∼10%.

## 2 RELATED WORK

**Counterfactual Explanation Techniques.** We categorize prior work on CF explanation techniques into *non-parametric methods* (Wachter et al., 2017; Ustun et al., 2019; Mothilal et al., 2020; Van Looveren & Klaise, 2019; Karimi et al., 2021; Upadhyay et al., 2021; Verma et al., 2020; Karimi et al., 2020), which aim to find recourses without involving parameterized models, and *parametric methods* (Pawelczyk et al., 2020b; Yang et al., 2021; Mahajan et al., 2019; Guo et al., 2021), which adopt parametric models (e.g., a neural network model) to generate recourses. In particular, our work is most closely related to CounterNet (Guo et al., 2021), which unlike post-hoc methods, jointly trains the predictive model and a CF explanation generator. *However, all aforementioned CF explanation techniques (including CounterNet) do not optimize for robustness against adversarial model shifts; instead, we devise a novel adversarial training approach to generate robust recourses.*

**Robustness in Counterfactual Explanations.** Our method is closely related to the model shift problem in algorithmic recourse (Rawal et al., 2020), i.e., how to ensure that the generated recourse is robust to shifts in the underlying predictive model. However, existing approaches (Upadhyay et al., 2021; Nguyen et al., 2022) rely on simplifying assumptions: (i) Upadhyay et al. (2021) propose ROAR which relies on a locally linear approximation (via LIME (Ribeiro et al., 2016)) to construct a shifted model, which is known to suffer from inconsistency (Slack et al., 2020; Alvarez-Melis & Jaakkola, 2018) and unfaithfulness issues (Laugel et al., 2018; Rudin, 2019). (ii) Similarly, Nguyen et al. (2022) propose RBR which assumes that kernel density estimators can approximate the underlying data distribution. In particular, RBR relies on Gaussian kernels for multivariate kernel density estimation, which suffers from the curse of dimensionality (Bellman, 1961; Crabbe, 2013; Nagler & Czado, 2016). In contrast, our work relaxes these assumptions by constructing adversarial shifted models via simulating the worst-case data shift, and conducting adversarial training for robust CF generation.

Orthogonal to our work, Pawelczyk et al. (2020a) analyze the *model multiplicity* problem, which studies the validity of recourses under different ML models trained on the *same* data, and Black et al. (2022) propose methods to ensure consistency under the model multiplicity setting. In addition, some prior work focuses on ensuring robustness to small perturbations in the feature space of generated recourses (Mishra et al., 2021; Fokkema et al., 2022; Dominguez-Olmedo et al., 2022).

**Adversarial training.** We also leverage adversarial robustness techniques, which are effective in defending ML models against adversarial examples (Goodfellow et al., 2014; Madry et al., 2017; Shafahi et al., 2019; Wong et al., 2019; Chen et al., 2022). In addition, recent works (Geiping et al., 2021; Gao et al., 2022) also leverage adversarial training to defend against data poisoning (Huang et al., 2020) and backdoor attacks (Saha et al., 2020). In general, adversarial training solves a bi-level (min-max) optimization problem. In our work, we formulate RoCourseNet's objective as a tri-level (min-max-min) optimization problem, which we can decompose into a game played between a model designer and a worst-case (hypothetical) adversary. The inner worst-case data and model shifts are assumed to be generated by a bi-level worst-case attacker. The defender trains a robust CF generator against this bi-level attacker by following an adversarial training procedure.

## 3 ROCOURSENET: END-TO-END ROBUST RECOURSE GENERATION

RoCourseNet is an end-to-end training framework for simultaneously generating accurate predictions and corresponding recourses (or CF explanations) that are robust to model shifts induced by shifts in the training dataset. At a high level, RoCourseNet jointly generates predictions and robust recourses by solving a tri-level optimization problem, which consists of two sub-problems: (i) a bi-level (max-min) problem which simulates a worst-case attacker by finding an adversarially shifted model; and (ii) an outer minimization problem that simulates a defender who wants to generate accurate predictions and robust recourses against this worst-case bi-level attacker.

We describe the RoCourseNet framework in two stages. First, we discuss the attacker's problem: (i) we propose a novel bi-level attacker problem to find the worst-case data shift that leads to an adversarially shifted ML model; and (ii) we propose a novel Virtual Data Shift (VDS) algorithm for solving this bi-level attacker problem. Second, we discuss the defender's problem: (i) we derive a novel tri-level learning problem based on the attacker's bi-level problem; and (ii) we propose the RoCourseNet training framework for optimizing this tri-level optimization problem, which leads to the simultaneous generation of accurate predictions and robust recourses.

## 3.1 VIRTUAL DATA SHIFT: CONSTRUCTING WORST-CASE DATA SHIFTS

We define a predictive model $f : \mathcal{X} \to \mathcal{Y}$. Let $\mathcal{D} = \{(x_i, y_i) \,|\, i \in \{1, \ldots, N\}\}$ represent our training dataset containing $N$ points. We denote $f(x, \theta)$ as the prediction generated by predictive model $f$ on point $x$, parameterized by $\theta$. Next, we denote $\theta$ and $\theta'$ as parameters of an (original) ML model $f(.; \theta)$ and its shifted counterpart $f(.; \theta')$, respectively. Also, let $x^{\mathrm{cf}}$ denote a CF explanation (or recourse) for input point $x$. Finally, a recourse $x^{\mathrm{cf}}$ is *valid* iff it gets an opposite prediction from the original data point $x$, i.e., $f(x^{\mathrm{cf}}; \theta) = 1 - f(x, \theta)$. On the other hand, a recourse $x^{\mathrm{cf}}$ is *robustly valid* w.r.t. a shifted model $f(.; \theta')$ iff $x^{\mathrm{cf}}$ gets an opposite prediction from the shifted model (as compared to the prediction received by $x$ on the original model), i.e., $f(x^{\mathrm{cf}}; \theta') = 1 - f(x; \theta)$. This definition aligns with the notion of robustness to model shifts in the literature (Upadhyay et al., 2021). Finally, $\mathcal{L}(., .)$ represents a loss function formulation, e.g., binary cross-entropy, mean squared error, etc.

**Model Shift as an optimization problem.** To motivate the need of introducing our bi-level attacker problem, we first discuss an optimization problem for a worst-case attacker which directly perturbs model parameters to find an adversarially shifted model (denoted by $f(.; \theta'_{adv})$). The goal of the attacker is to find an adversarially shifted model which minimizes the robustness of the generated recourses. More formally, given our training dataset $\mathcal{D}$ and a CF explanation method (that can generate recourses $x^{\mathrm{cf}}$ for each $(x, y) \in \mathcal{D}$), the attacker's problem aims to find the worst-case shifted model $f(.; \theta'_{adv})$ which minimizes the robust validity of the generated recourses, i.e., $f(x^{\mathrm{cf}}; \theta'_{adv}) \neq 1 - f(x; \theta)$. We can find an adversarial shifted model by solving Equation 1.

$$\theta'_{adv} = \mathrm{argmax}_{\theta' \in \mathcal{F}} \; \frac{1}{N} \sum\nolimits_{(x_i, y_i) \in \mathcal{D}} \left[ \mathcal{L}\Big( f\left(x_i^{\mathrm{cf}}; \theta'\right), 1 - f\left(x_i; \theta\right) \Big) \right] \tag{1}$$

where $x^{\mathrm{cf}}$ correspond to CF explanations of input $x$ produced by a CF generator, and $\mathcal{F} = \{\theta' \,|\, \theta + \delta_f\}$ denotes a plausible set of the parameters of all possible shifted models.

Unfortunately, it is non-trivial to construct a plausible model set $\mathcal{F}$ by directly perturbing the ML model's parameters $\theta$, especially when $f(\cdot; \theta)$ is represented using a neural network. Unlike a linear model, quantifying the importance of neurons (to get neuron-specific weights) is challenging (Leino et al., 2018; Dhamdhere et al., 2018), which leads to difficulty in applying weight perturbations. To overcome these challenges, prior work (Upadhyay et al., 2021) adopts a simplified linear model to approximate the target model, and perturbs this linear model accordingly. Unfortunately, this simplified local linear model introduces approximation errors into the system, which leads to poor performance (as shown in Section 4). Instead of directly perturbing the model's weights, we explicitly consider a worst-case data shift, which then leads to an adversarial model shift.

**Data Shift as a bilevel optimization problem.** We identify distributional data shifts as the fundamental cause of non-robust recourses. Unlike prior work, we propose a bi-level optimization problem for the attacker which explicitly connects shifts in the underlying training data to corresponding shifts in the ML model parameters. Specifically, in response to a shift in the training data (from $\mathcal{D}$ to $\mathcal{D}_{\mathrm{shifted}}$), defenders update (or shift) their predictive model by optimizing the prediction loss:

$$\theta'_{opt} = \mathrm{argmin}_{\theta'} \; \frac{1}{N} \sum\nolimits_{(x_i, y_i) \in \mathcal{D}_{\mathrm{shifted}}} \left[ \mathcal{L}\Big( f(x_i; \theta'), y_i \Big) \right]. \tag{2}$$

Crucially, this updated ML model $f(\cdot, \theta'_{opt})$ (caused by the shifted data $\mathcal{D}_{\mathrm{shifted}}$) is the key reason which leads to the non-robustness of recourses (i.e., $f(x^{\mathrm{cf}}; \theta'_{opt}) \neq 1 - f(x; \theta)$). Therefore, to generate robust recourses, we optimize against an adversary who creates a worst-case shifted dataset $\mathcal{D}^*_{\mathrm{shifted}}$, such that the correspondingly updated model (found by solving Eq. 2) minimizes the robust

validity of CF examples $x^{\text{cf}}$. Then, this data shift problem can be formulated as a bi-level problem:

$$\boldsymbol{\delta}^* = \operatorname*{argmax}_{\boldsymbol{\delta}, \forall \delta_i \in \Delta} \frac{1}{N} \sum_{(x_i, y_i) \in \mathcal{D}} \left[ \mathcal{L}\Big( f(x_i^{\text{cf}}; \theta'_{opt}(\boldsymbol{\delta})), 1 - f(x_i; \theta) \Big) \right]$$

$$s.t., \theta'_{opt}(\boldsymbol{\delta}) = \operatorname{argmin}_{\theta'} \frac{1}{N} \sum_{(x_i, y_i) \in \mathcal{D}} \left[ \mathcal{L}\Big( f(x_i + \delta_i; \theta'), y_i \Big) \right]. \quad (3)$$

where $\delta_i \in \Delta$ denotes the data shift for a single data point $x_i \in \mathcal{D}$, and $\boldsymbol{\delta} = \{ \delta_i \mid \forall (x_i, y_i) \in \mathcal{D} \}$ denotes the data shift across all data points in the entire dataset. We define $\Delta$ as the $l_\infty$-norm ball $\Delta = \{ \delta \in \mathbb{R}^n \mid \|\delta\|_\infty \leq \epsilon \}$ (see $l_2$-norm based definitions in Appendix). Intuitively, the outer problem minimizes the robust validity to construct the worst-case data shift $\boldsymbol{\delta}^*$, and the inner problem learns a shifted model $f(.; \theta'_{opt})$ on the shifted dataset $\mathcal{D}^*_{\text{shifted}}$. Once we get the optimal $\boldsymbol{\delta}^* = \{ \delta_1^*, \delta_2^*, \ldots, \delta_N^* \}$ by solving Equation 3, the worst-case shifted dataset $\mathcal{D}^*_{\text{shifted}} = \{ (x_i + \delta_i^*, y_i) \mid \forall (x_i, y_i) \in \mathcal{D} \}$.

**Virtual Data Shift (VDS).** Unfortunately, solving Eq. 3 is computationally intractable due to its nested structure. To approximate this bi-level problem, we devise *Virtual Data Shift (VDS)* (Algorithm 1), a gradient-based algorithm with an unrolling optimization pipeline. At a high level, VDS iteratively approximates the inner problem by unrolling $K$-steps of gradient descent for each outer optimization step. Similar unrolling optimization pipelines are commonly adopted in many ML problems with a bi-level formulation (Shaban et al., 2019; Gu et al., 2022), e.g., meta-learning (Finn et al., 2017), hyperparameter search (Maclaurin et al., 2015), and poisoning attacks (Huang et al., 2020).

Algorithm 1 layouts the *VDS* algorithm which outputs the worst-case data shift $\boldsymbol{\delta}^*$, and the corresponding shifted model $f(.; \theta'_{opt})$. VDS makes two design choices. First, it uniformly randomizes the data shift $\boldsymbol{\delta} \sim \mathcal{U}(-\epsilon, +\epsilon)$, where $\boldsymbol{\delta} = \{ \delta_1, \ldots, \delta_N \}$ (Line 3), following practices of Wong et al. (2019). Uniform randomization is critical to adversarial model performance as it increases the smoothness of the objective function, leading to improved convergence of gradient-based algorithms (Chen et al., 2022). Then, we iteratively solve this bi-level optimization problem via $T$ outer attack steps. At each step, we first update the predictor $f(.; \theta')$ using the shifted data $\mathbf{x} + \boldsymbol{\delta}$ via $K$ unrolling steps of gradient descent, where $\mathbf{x} = \{ x_1, \ldots, x_N \}$ (Line 6). Next, similar to the fast sign gradient method (FSGM) (Goodfellow et al., 2014), we maximize the adversarial loss and project $\boldsymbol{\delta}$ into the feasible region $\Delta$ (i.e., $l_\infty$ norm ball; Line 8-9). Crucially, when calculating the gradient of adversarial loss (outer problem) with respect to data shift $\boldsymbol{\delta}$ (Line 8), we look ahead in the inner problem for a few forward steps and then back-propagate to the initial unrolling step. We do this because we use $K$ unrolling steps of gradient descent, as opposed to full-blown gradient descent till convergence. Note that the gradient w.r.t. $\delta$ depends on $\theta(\delta)$, where $\theta(\delta)$ is a function resulting from LINE 5-7.

---

**Algorithm 1** Virtual Data Shift (VDS)

---

1: **Hyperparameters:** learning rates $\eta$, step size $\alpha$, # of attacker steps $T$, # of unrolling steps $K$
2: **Input:** model weights $\theta$, perturbation constraints $\epsilon$, batch $B = (\mathbf{x}, y)$, CF examples $\mathbf{x}^{\text{cf}}$
3: **Initialize:** virtual shifted model weights $\theta' = \theta$, $\boldsymbol{\delta} \sim \mathcal{U}(-\epsilon, +\epsilon)$
4: **for** $i = 1 \rightarrow T$ steps **do**
5:     **for** $k = 1 \rightarrow K$ unroll steps **do**
6:         $\theta' \leftarrow \theta' - \eta \cdot \nabla_{\theta'} \mathcal{L}\left( f(\mathbf{x} + \boldsymbol{\delta}); \theta' \right), y)$
7:     **end for**
8:     $\boldsymbol{\delta} \leftarrow \boldsymbol{\delta} + \alpha \cdot sign\left( \nabla_\delta \mathcal{L}\left( f(\mathbf{x}^{\text{cf}}; \theta'), 1 - f(\mathbf{x}; \theta) \right) \right)$
9:     Project $\boldsymbol{\delta}$ onto the $l_\infty$-norm ball.
10: **end for**
11: **return** $\theta', \boldsymbol{\delta}$

---

### 3.2 BLOCK-WISE COORDINATE DESCENT WITH ADVERSARIAL TRAINING

**Choice of CF Explanation Technique.** Note that our bi-level attacker formulation assumes that CF explanations $x_i^{\text{cf}}$ for all data points $x_i$ are provided as input to the VDS algorithm. Thus, a key design choice inside the RoCourseNet framework is the selection of an appropriate CF explanation technique, which can be used to generate recourses for input data points in Algorithm 1.

As mentioned in Section 2, most existing CF explanation techniques belong to the post-hoc explanation paradigm, which makes them unsuitable for use inside the RoCourseNet framework for two

reasons: (ii) *misaligned motivations*: post-hoc CF explanation methods are mainly designed for use with proprietary black-box ML models whose training data and model weights are not available; instead, the VDS algorithm relies on having access to the training data. Thus, the motivations and use-cases of VDS and post-hoc methods are misaligned. (ii) the post-hoc paradigm is overly limiting in many real-world scenarios. With the advent of data regulations that enshrine the "*Right to Explanation*" (e.g., EU-GDPR (Wachter et al., 2017)), service providers are required by law to communicate both the decision outcome (i.e., the ML model's prediction) and its actionable implications (i.e., a recourse for this prediction) to an end-user. In these scenarios, the post-hoc assumption is overly limiting, as service providers can build specialized CF explanation techniques that can leverage the knowledge of their particular ML model to generate higher-quality recourses.

Motivated by these reasons, we choose CounterNet (Guo et al., 2021) as our CF explanation model of choice inside the RoCourseNet framework, as that is an end-to-end approach that departs from the post-hoc explanation paradigm by jointly training accurate predictions and recourses. In fact, Guo et al. (2021) show that CounterNet is able to better balance the cost-invalidity trade-off (Rawal et al., 2020) as compared to state-of-the-art post-hoc approaches.

**RoCourseNet Objective Function.** We now describe how our bi-level attacker problem is plugged into an outer minimization problem to formulate a tri-level problem, which represents RoCourseNet's objective function. Inspired by Guo et al. (2021), RoCourseNet has three objectives: (i) high *predictive accuracy* - we expect RoCourseNet to output accurate predictions; (ii) high *robust validity* - we expect that generated recourses in RoCourseNet are robustly valid on shifted models[2]; (iii) low *proximity* - we desire minimal changes required to modify input instance $x$ to corresponding recourse $x^{\text{cf}}$. Based on these objectives, RoCourseNet solves the following min-max-min optimization problem to find parameters for its predictive model $f(.;\theta)$ and recourse generator $g(.;\theta_g)$. Note that similar to Guo et al. (2021), both $f(.;\theta)$ and $g(.;\theta_g)$ are represented using neural networks.

$$
\underset{\theta,\theta_g}{\operatorname{argmin}} \frac{1}{N} \sum_{(x_i,y_i)\in\mathcal{D}} \left[ \lambda_1 \cdot \underbrace{\mathcal{L}\Big(f(x_i;\theta),y_i\Big)}_{\text{Prediction Loss }(L_1)} + \lambda_3 \cdot \underbrace{\mathcal{L}\Big(x_i,x_i^{\text{cf}}\Big)}_{\text{Proximity Loss }(L_3)} \right]
$$

$$
+ \underset{\boldsymbol{\delta},\forall\delta_i\in\Delta}{\max} \frac{1}{N} \sum_{(x_i,y_i)\in\mathcal{D}} \left[ \lambda_2 \cdot \underbrace{\mathcal{L}\Big(f\left(x_i^{\text{cf}};\theta'_{opt}(\boldsymbol{\delta})\right),1-f\left(x_i;\theta\right)\Big)}_{\text{Robust Validity Loss }(L_2)} \right] \quad (4)
$$

$$
s.t \ \theta'_{opt}(\boldsymbol{\delta}) = \operatorname{argmin}_{\theta'} \frac{1}{N} \sum_{(x_i,y_i)\in\mathcal{D}} \left[ \mathcal{L}\Big(f(x_i+\delta_i;\theta'),y_i\Big) \right], \ x_i^{\text{cf}} = g(x_i;\theta_g).
$$

**RoCourseNet Training.** A common practice in solving a min-max formulation is to first solve the inner maximization problem, and then solve the outer minimization problem (Madry et al., 2017). Thus, we solve Eq. 4 as follows (see Algorithm 2): (i) To solve the inner bi-level problem, we find the worst-case model shift by solving Eq. 3 using *VDS* (Algorithm 1). (ii) To solve the outer minimization problem, we adopt block-wise coordinate descent optimization by distributing gradient descent backpropagation on the objective function into two stages (as suggested in (Guo et al., 2021)) – at stage one, we optimize for predictive accuracy, i.e., $L_1$

---

**Algorithm 2** Tri-level Robust CF Training
1: **Hyperparameters:** learning rates $\eta$, # of epochs $N$, maximum perturbation $E$
2: **Input:** dataset $(x,y)\in\mathcal{D}$
3: **Initialize:** $\theta$.
4: **for** epoch $= 1 \to N$ **do**
5:     $\epsilon = E \cdot \text{epoch}/N$     ▷ Linearly schedule $\epsilon$.
6:     **for** each minibach $B$ in $\mathcal{D}$ **do**
7:         $\theta \leftarrow \theta - \eta \cdot \nabla_\theta L_1$
8:         $\theta',\delta \leftarrow \text{VDS}(B, x^{\text{cf}}, \theta, \epsilon)$
9:         $\theta_g \leftarrow \theta_g - \eta \cdot \nabla_{\theta_g} (\lambda_2 \cdot L_2 + \lambda_3 \cdot L_3)$
10:     **end for**
11: **end for**

---

in Eq. 4 (Line 7), and at stage two, optimize the quality of CF explanations, i.e., $\lambda_2 \cdot L_2 + \lambda_3 \cdot L_3$ in Eq. 4 (Line 9). Note that in Line 9, the gradient of $\lambda_2 \cdot L_2 + \lambda_3 \cdot L_3$ is calculated using the adversarially shifted model parameters $\theta'$ found by VDS in Line 8. Guo et al. (2021) show that this block-wise coordinate descent algorithm efficiently optimizes the outer minimization problem by alleviating the problem of divergent gradients.

---

[2]Note that robust validity also ensures validity on the original predictive model (i.e., $f(x;\theta) = 1 - f(x^{\text{cf}};\theta)$), as the worst-case model shift case encompasses the unshifted model case.

In addition, we linearly increase the perturbation constraints $\epsilon$ for improved convergence of the training of our robust CF generator. Intuitively, linearly increasing $\epsilon$ values corresponds to increasingly strong (hypothetical) adversaries. Prior work in curriculum adversarial training (Cai et al., 2018; Wang et al., 2019) suggests that adaptively adjusting the strength of the adversary improves the convergence of adversarial training (we verify this in Section 4).

# 4 EXPERIMENTAL EVALUATION

**Baselines.** Since, to our knowledge, RoCourseNet is the first method which optimizes for an end-to-end model for generating predictions and robust recourses, there exist no previous approaches which we can directly compare against. Nevertheless, we compare RoCourseNet with two post-hoc robust recourse generation methods: (i) ROAR-LIME (Upadhyay et al., 2021); and (ii) RBR (Nguyen et al., 2022). In addition, to remain consistent with baselines established in prior work (Upadhyay et al., 2021; Nguyen et al., 2022), we also compare against VANILLACF (Wachter et al., 2017), a popular post-hoc method. Finally, we also compare against COUNTERNET (Guo et al., 2021) since our approach is inspired by their end-to-end paradigm of generating recourses.

Other than *CounterNet*, all remaining baselines require a trained ML model as an input. Similar to (Guo et al., 2021), we train a neural network model and use it as the base ML model for all baselines. For each dataset, we separately tune hyperparameters using grid search (see Appendix for details).

**Datasets.** To remain consistent with prior work on robust recourses (Upadhyay et al., 2021; Nguyen et al., 2022), we evaluate RoCourseNet on three benchmarked real-world datasets. (i) *Loan* (Li et al., 2018), which captures *temporal shifts* in loan application records. It predicts whether a business defaulted on a loan (Y=1) or not (Y=0). This large-sized dataset consists of 449,152 loan approval records across U.S. during the years 1994 to 2009 (i.e., $\mathcal{D}_{\text{all}} = \{\mathcal{D}_1, \mathcal{D}_2, ..., \mathcal{D}_k\}$, where each subset $\mathcal{D}_i$ corresponds to a particular year, and $k = 16$ is the total number of years). (ii) *German Credit* (Asuncion & Newman, 2007), which captures data correction shifts. It predicts whether the credit score of a customer is good (Y=1) or bad (Y=0). This dataset has 2,000 data points with two versions; each version contains 1,000 data points (i.e., $\mathcal{D}_{\text{all}} = \{\mathcal{D}_1, \mathcal{D}_2\}$, where $\mathcal{D}_1, \mathcal{D}_2$ corresponds to the original and corrected datasets, respectively). (iii) Finally, we use the *Student* dataset (Cortez & Silva, 2008), which captures geospatial shifts. It predicts whether a student will pass (Y=1) or fail (Y=0) the final exam. It contains 649 student records in two locations (i.e., $\mathcal{D}_{\text{all}} = \{\mathcal{D}_1, \mathcal{D}_2\}$, where each subset $\mathcal{D}_i$ corresponds to a particular location).

**Evaluation Procedure & Metrics.** Each dataset is partitioned into $k$ subsets $\mathcal{D}_{\text{all}} = \{\mathcal{D}_1, \mathcal{D}_2, ..., \mathcal{D}_k\}$. This partitioning enables us to create $k$ original datasets $\mathcal{D}_i \in \mathcal{D}_{\text{all}}$, and $k$ corresponding shifted datasets, $\mathcal{D}_{\text{shifted},i} = \mathcal{D}_{\text{all}} \setminus \mathcal{D}_i$, where each shifted dataset $\mathcal{D}_{\text{shifted},i}$ contains all subsets of the dataset $\mathcal{D}_{\text{all}}$ except for the original dataset $\mathcal{D}_i$. Next, we do a train/test split on each $\mathcal{D}_i \in \mathcal{D}_{\text{all}}$ (i.e., $\mathcal{D}_i = \{\mathcal{D}_i^{\text{train}}, \mathcal{D}_i^{\text{test}}\}$). We train a separate model (i.e., the entire models for *RoCourseNet* and *CounterNet*, and predictive models for other baselines) on each train set $\mathcal{D}_i^{\text{train}}$, $\forall i \in \{1, ..., k\}$. Then, we use the model trained on $\mathcal{D}_i^{\text{train}}$ to generate recourses on the hold-out sets $\mathcal{D}_i^{\text{test}}$, $\forall i \in \{1, ..., k\}$. Finally, we evaluate the robustness (against the model shift) of recourses generated on $\mathcal{D}_i^{\text{test}}$, $\forall i \in \{1, ..., k\}$ as follows: for each recourse $x^{\text{cf}}$ (corresponding to an input instance $x$ in $\mathcal{D}_i^{\text{test}}$), we evaluate its robust validity by measuring the fraction of shifted models (i.e., $k - 1$ models trained on all shifted training sets) on which $x^{\text{cf}}$ remains *robustly valid* (see definitions in Section 3.1).

Finally, we use three metrics to evaluate a CF explanation: (i) *Validity* is defined as the fraction of valid CF examples on the original model $f(.; \theta)$. (ii) *Robust validity* is defined as the fraction of robustly valid CF examples on a *shifted* predictive model $f(.; \theta')$. We calculate the robust validity on all possible shifted models (as described above). (iii) *Proximity* is defined as the $l_1$ distance between the input and the CF example. We report the averaged results across all subsets $\mathcal{D}_{all}$ (see Table 1).

**Validity & Robust Validity.** Table 1 compares the validity and robust validity achieved by RoCourseNet and other baselines. RoCourseNet achieves the highest *robust validity* on each dataset - it outperforms ROAR-LIME by 10% (the next best performing baseline), and consistently achieves at least 96.5% robust validity. This illustrates RoCourseNet's effectiveness at generating highly robust recourses. Also, RoCourseNet achieves the highest *validity* on each dataset. This result shows that optimizing the worst-case shifted model (i.e., $L_2$ in Eq. 4) is sufficient to achieve high validity, without the need to explicitly optimize for an additional validity loss on the original model.

Table 1: **Evaluation of recourse robustness under model shift.** It is desirable for recourse methods to have *low* proximity (prox.) with *high* validity (Val.) and *high* robust validity (Rob-Val.).

| Methods | Loan | | | German Credit | | | Student | | |
|---|---|---|---|---|---|---|---|---|---|
| | Prox. | Val. | Rob-Val. | Prox. | Val. | Rob-Val. | Prox. | Val. | Rob-Val. |
| VANILLACF | 7.390 ± 1.860 | 0.942 ± 0.026 | 0.885 ± 0.121 | **4.635** ± **0.197** | 0.940 ± 0.011 | 0.772 ± 0.008 | 15.236 ± 0.383 | 0.915 ± 0.056 | 0.673 ± 0.006 |
| COUNTERNET | 6.746 ± 0.723 | 0.964 ± 0.085 | 0.639 ± 0.222 | 5.719 ± 0.130 | 0.960 ± 0.028 | 0.706 ± 0.074 | 18.619 ± 0.131 | 0.967 ± 0.033 | 0.859 ± 0.071 |
| ROAR-LIME | 7.648 ± 1.951 | 0.934 ± 0.024 | 0.918 ± 0.066 | 4.862 ± 0.117 | 0.910 ± 0.0255 | 0.792 ± 0.052 | **11.931** ± **1.396** | 0.967 ± 0.026 | 0.820 ± 0.075 |
| RBR | 11.71 ±1.633 | 0.824 ± 0.130 | 0.821 ± 0.132 | 6.005 ± 2.099 | 0.524 ± 0.148 | 0.586 ± 0.046 | 26.255 ± 3.089 | 0.660 ± 0.014 | 0.611 ± 0.073 |
| ROCOURSENET | **6.611** ± **0.418** | **0.996** ± **0.002** | **0.969** ± **0.106** | 6.903 ± 0.250 | **0.978** ± **0.002** | **0.964** ± **0.008** | 16.508 ± 0.281 | **1.000** ± **0.000** | **1.000** ± **0.000** |

**Proximity.** Table 1 compares the proximity achieved by RoCourseNet and baselines. In particular, RoCourseNet performs exceedingly well on the Loan application dataset (our largest dataset with ~450k data points), as it is the second-best method in terms of proximity (just behind CounterNet), and outperforms all our post-hoc baseline methods (RBR, ROAR-LIME, and VanillaCF). Perhaps understandably, RoCourseNet achieves poorer proximity on the German Credit and Student dataset, given that the limited size of these datasets (less than 1000 data points) precludes efficient training.

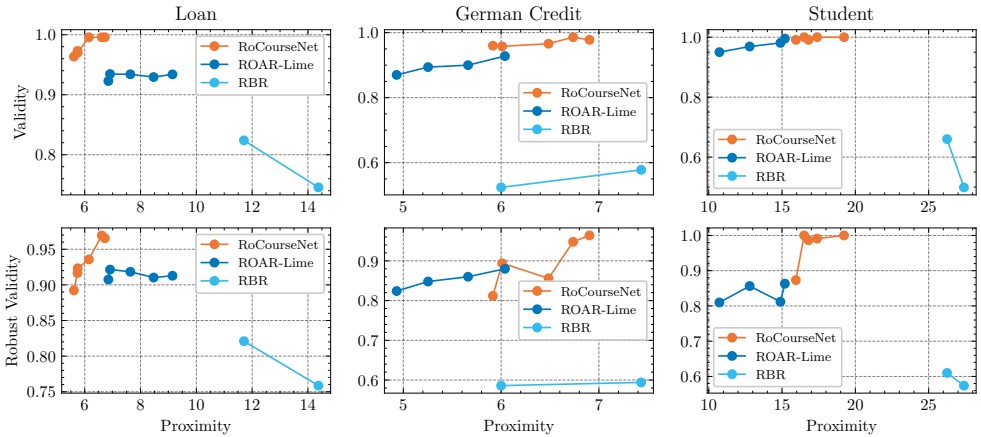

Figure 2: Pareto frontiers of the cost-invalidity trade-off.

**Cost-Validity Trade-Off.** We compare RoCourseNet, ROAR-LIME, and RBR (three recourse methods explicitly optimizing for distributional shift) in their trade-off between the cost (measured by proximity) and their original and robust validity (Rawal et al., 2020). For each method, we plot the Pareto frontier of the cost-validity trade-off as follows: (i) For RBR, we obtain the Pareto frontier by varying the ambiguity sizes $\epsilon_1$, $\epsilon_2$ (i.e., hyperparameters of RBR Nguyen et al. (2022)); (ii) For ROAR-LIME and CounterNet, we obtain the Pareto frontier by varying the trade-off hyperparameter $\lambda$ that controls the proximity loss term in their respective loss functions (e.g., $\lambda_3$ in Equation 4). Figure 2 shows that on the large-sized Loan dataset, RoCourseNet's Pareto frontiers either dominate or are comparable to frontiers achieved by

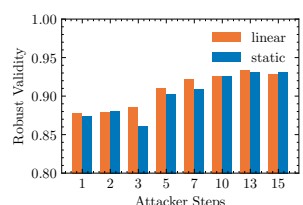

Figure 4: Impact of epsilon scheduler on the robustness.

ROAR-LIME and RBR. On the other hand, we observe a clear trade-off on the German Credit and Student datasets, where RoCourseNet (and other methods) can increase their normal and robust validity, but only at the cost of a poorer proximity score. This is consistent with prior literature, which shows that proximity needs to be sacrificed to achieve higher validity (Nguyen et al., 2022).

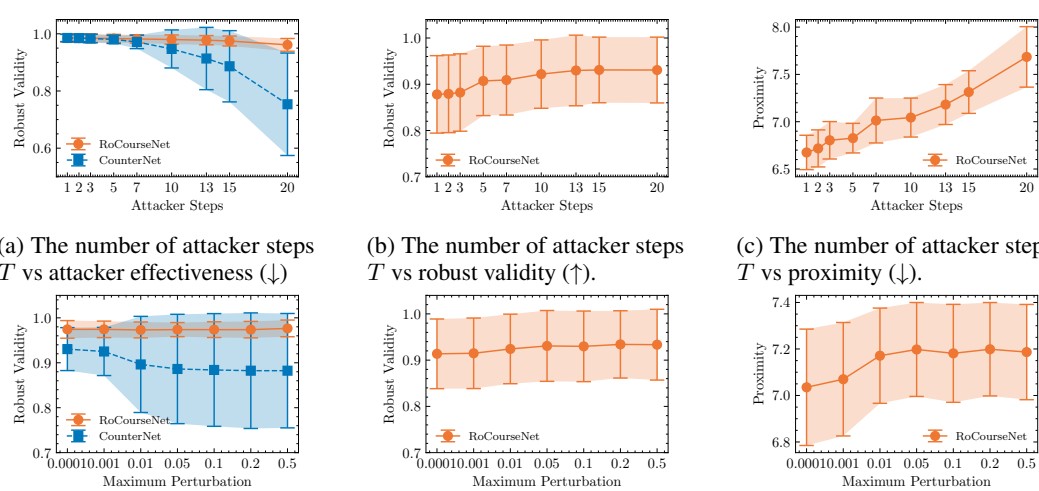

Figure 3: The impact of the number of attacker steps $T$ (3a-3c) and max-perturbation $E$ (3d-3f) to robustness on the *Loan* dataset ($\uparrow$ or $\downarrow$ means that higher or lower value is preferable, respectively).

**Evaluating VDS Attacker.** We first demonstrate the effectiveness of VDS on solving the inner maximization problem in Eq. 4, i.e., how often VDS succeeds in finding an adversarial shifted model $f(.; \theta')$, such that the generated recourses $x^{\mathrm{cf}}$ are not robustly valid. To evaluate the performance of VDS, we apply Algorithm 1 to find shifted model parameters $\theta'$, and compute the robust validity of all hold-out test-sets with respect to this shifted model.

Figure 3a and 3d compares the effectiveness of attacking RoCourseNet and CounterNet via the VDS algorithm, which highlights three important findings: (i) First, the VDS algorithm is effective in finding a shifted model which invalidates a given recourse - the average robust validity drops to 74.8% when the attacker steps $T = 20$, as compared to 99.8% validity on the original model. (ii) Figure 3a shows that when $T$ is increased, the robust validity of both CounterNet and RoCourseNet is degraded, which indicates that increasing attack steps improves the effectiveness of solving the bi-level problem in Eq. 3. Similarly, Figure 3d shows that increasing $E$ also improves effectiveness of solving Eq.3. (iii) Finally, RoCourseNet is more robust than CounterNet when attacked by the VDS algorithm, as RoCourseNet vastly outperforms CounterNet in robust validity (e.g., $\sim$28%, $\sim$10% improved robust validity when $T = 20$, $E = 0.5$ in Figure 3a and 3d, respectively).

**Understanding the Tri-level Robust CF Training.** We further analyze our tri-level robust training procedure. First, we observe that a stronger attacker (i.e., more effective in solving Eq. 3) contributes to the training of a more robust CF generator. In Figure 3b, we observe that increasing $T$ (which results in a stronger attacker as shown in Figure 3a) leads to improved robust validity, which indicates a more robust CF generator. Similarly, Figure 3e illustrates that increasing $E$ values leads to improved robustness of counterfactual validity. These results show that having an appropriately strong attacker (i.e., effectively optimizing Eq. 3) is crucial to train for a robust CF generator.

**Epsilon Scheduler.** Figure 4 illustrates the importance of linearly scheduling $\epsilon$ values inside VDS (see Appendix for experiments with non-linear schedulings). By linearly increasing $\epsilon$, we observe $\sim$0.88% improved robust validity (on average) compared to using a static $\epsilon$ during the entire adversarial training. This shows that this curriculum training strategy can boost the robustness of recourses.

## 5 CONCLUSION

We present *RoCourseNet*, an end-to-end training framework to generate predictions and robust CF explanations. We formulate this robust end-to-end training as a min-max-min optimization problem, and leverage novel adversarial training techniques to effectively solve this complicated tri-level problem. Empirical results show that RoCourseNet outperforms state-of-the-art baselines in terms of robust validity, and balances the cost-validity trade-off significantly better than baselines.

## 6 ETHICS & REPRODUCIBLILITY STATEMENT

**Ethics Statement.** RoCoursenet takes into account of potential model shift in the real-world deployment. However, before deploying RoCourseNet to a real-world system, one must be aware of the possibility for RoCourseNet to reinforce pre-existing societal prejudices among end users. A temporary workaround is to have trained human operators to communicate the CF explanations in a respectful manner. In the long term, more quantitative and qualitative research will be required to fully understand RoCourseNet's societal impact.

**Reproducibility Statement.** We release the code through an anonymous repository in supporting the reproduction of this work ((https://github.com/bkghz-orange-blue/counternet_adv). In this anonymous repository, we also offer the dataset that was utilized for this paper's evaluation. We also include the options for hyperparameters in the appendix A. Appendix A contains a thorough description of our experimental implementation.

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

# A    IMPLEMENTATION DETAILS

Here we provide implementation details of RoCourseNet and three baseline methods on three datasets listed in Section 4. The code can be found through this anonymous repository (`https://github.com/bkghz-orange-blue/counternet_adv`).

**Feature Engineering.** We follow the feature engineering procedure of CounterNet (Guo et al., 2021). Specifically, for continuous features, we scale all feature values into the $[0, 1]$ range. To handle the categorical features, we customize model architecture for each dataset. First, we transform the categorical features into numerical representations via one-hot encoding. In addition, for each categorical feature, we add a softmax layer after the final output layer in the CF generator, which ensures that the generated CF examples respect the one-hot encoding format.

**Hyperparameters.** For all three datasets, we train the model for up to 50 epochs with Adam. We set dropout rate to 0.3 to prevent overfitting. We use $T = 7$ and $E = 0.1$ to report results in Table 1, and report the impact of attacker steps $T$ and maximum perturbation $E$ to robustness in Figure 3. We use $K = 2$ unrolling steps (same as Huang et al. (2020)) with the step size $\alpha = 2.5 \times \delta/T$ (based on Madry et al. (2017)) for solving the bi-level problem in Equation 3 (via VDS). In addition, Table 2 reports the hyperparameters chosen for each dataset, and Table 3 spefices the architecture used for each dataset.

Table 2: Hyperparameters setting for each dataset.

| Dataset | Learning Rate | $\eta$ | Batch Size | $\lambda_1$ | $\lambda_2$ | $\lambda_3$ |
|---|---|---|---|---|---|---|
| **Loan** | 0.003 | 0.03 | 128 | 1.0 | 0.2 | 0.1 |
| **German Credit** | 0.003 | 0.03 | 256 | 1.0 | 1.0 | 0.1 |
| **Student** | 0.01 | 0.01 | 128 | 1.0 | 0.2 | 0.1 |

Table 3: Architecture specification of RoCourseNet for each dataset.

| Dataset | Encoder Dims | Predictor Dims | CF Generator Dims |
|---|---|---|---|
| **Loan** | [110,200,10] | [10, 10] | [10, 10] |
| **German Credit** | [19, 100,10] | [10, 20] | [10, 20] |
| **Student** | [83,50,10] | [10, 10] | [10, 50] |

**Software and Hardware Specifications.** We use Python (v3.7) with Pytorch (v1.82), Pytorch Lightning (v1.10), numpy (v1.19.3), pandas (v1.1.1), scikit-learn (v0.23.2) and higher (v0.2.1) Grefenstette et al. (2019) for the implementations. All our experiments were run on a Debian-10 Linux-based Deep Learning Image on the Google Cloud Platform. The RoCourseNet and baseline methods are trained (or optimized) on a 16-core Intel machine with 64 GB of RAM.

# B    ADDITIONAL EXPERIMENTAL ANALYSIS

## B.1    PREDICTIVE PERFORMANCE

We first show that, similar to CounterNet, the training of RoCourseNet does not come at the cost of degraded predictive accuracy. Table 4 & 5 compare RoCourseNet's predictive accuracy and AUC score against the base prediction model used by baselines. This table shows that RoCourseNet achieves competitive predictive performance – it achieves marginally better accuracy than the base model ($\sim 2\%$). Thus, we conclude that the joint training of RoCourseNet does not come at a cost of reduced predictive performance.

## B.2    HEURISTIC BASELINES

We provide two heuristic baseline methods to further illustrate the challenge of generating robust recourses under the distribution shift scenarios. *VanillaCF-Random* aims to generate robust recourse

Table 4: Predictive accuracy for each dataset.

| Dataset | Base Model | RoCourseNet |
|---|---|---|
| Loan | $0.886 \pm 0.036$ | $0.885 \pm 0.035$ |
| German Credit | $0.714 \pm 0.003$ | $0.742 \pm 0.014$ |
| Student | $0.914 \pm 0.028$ | $0.906 \pm 0.066$ |

Table 5: AUC score for each dataset.

| Dataset | Base Model | RoCourseNet |
|---|---|---|
| Loan | $0.897 \pm 0.026$ | $0.900 \pm 0.027$ |
| German Credit | $0.662 \pm 0.018$ | $0.729 \pm 0.010$ |
| Student | $0.913 \pm 0.012$ | $0.947 \pm 0.018$ |

by adding a small perturbation to input. In addition, *RoCourseNet-Random* optimizes for robust CF generator against a random perturbation attacker.

Table 6 compares heuristic baselines with RoCourseNet. Both baseline methods peroform significantly worse than RoCourseNet in validity, robust validity and proximity. This experiment further highlights the hardness of generating robust recourses as simple heuristics drastically underperform as compared to RoCourseNet.

Table 6: Heuristic baselines as compared to RoCourseNet on Loan dataset. Simple heuristic does not defend against distribution shift.

| Method | Validity | Robust Validity | Proximity |
|---|---|---|---|
| VANILLACF-RANDOM | $0.634 \pm 0.270$ | $0.510 \pm 0.251$ | $9.446 \pm 1.042$ |
| ROCOURSENET-RANDOM | $0.856 \pm 0.127$ | $0.856 \pm 0.127$ | $10.405 \pm 1.857$ |
| ROCOURSENET | $\mathbf{0.996 \pm 0.002}$ | $\mathbf{0.969 \pm 0.106}$ | $\mathbf{6.611 \pm 0.418}$ |

### B.3 SIMULATED DATA SHIFT

We conduct simulated experiments with covariant and label shift.

**Covariant Shift**    We simulate the covariant shift via this Bayesian network:

$$x_1 \sim \mathcal{N}(\mu_1, \sigma_1)$$
$$x_2 \sim \mathcal{N}(\mu_2, \sigma_2)$$
$$y = -x_2 + x_1^3 + \varepsilon, \ \varepsilon \sim \mathcal{N}(-0.1, 0.1)$$

where we set $\mu_1 = 0.5, \sigma_1 = 0.5, \mu_2 = 0, \sigma_2 = 0.3$ for $D_1$, and $\mu_1 = 0, \sigma_1 = 0.3, \mu_2 = 0.5, \sigma_2 = 0.5$ for $D_2$. Figure 5 illustrates this simulation dataset.

**Label Shift**    Similarly, we simulate the label shift via this Bayesian network:

$$y \sim binomial(p)$$
$$z = 2y - 1 + \varepsilon, \ \varepsilon \sim \mathcal{N}(0.1, 0.1)$$
$$x1 \sim \mathcal{N}(-z + z^3, 0.3)$$
$$x2 \sim \mathcal{N}(z + z^3 - 3y, 0.3)$$

where we set $p = 0.6$ for $D_1$, and $p = 0.3$ for $D_2$. Figure 6 illustrates this simulation dataset.

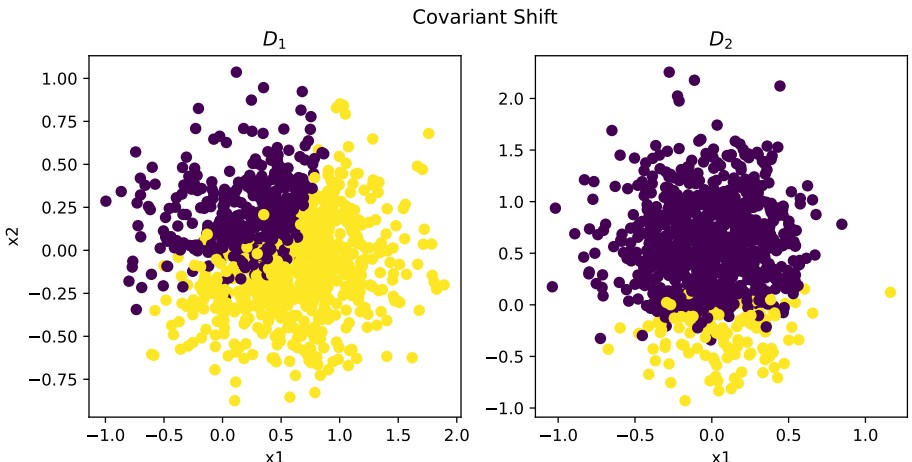

Figure 5: Illustration of covariant shift.

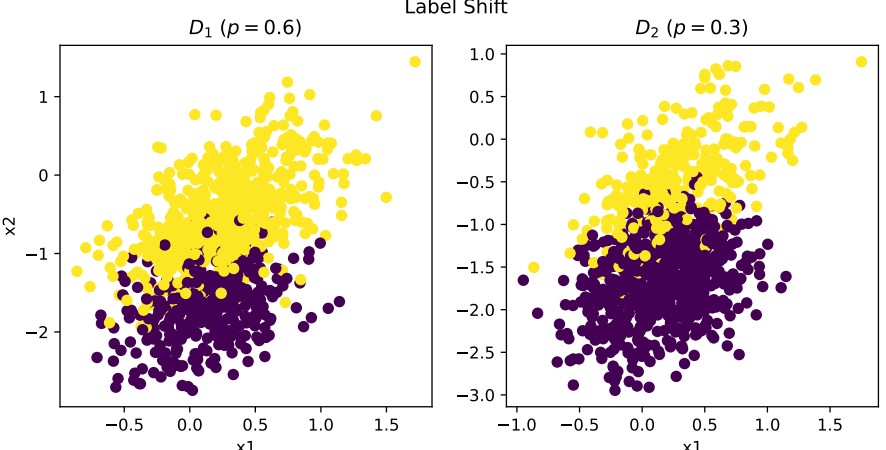

Figure 6: Illustration of label shift.

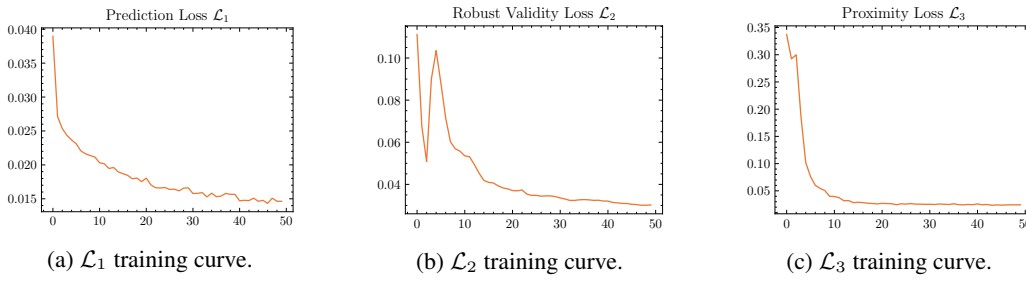

(a) $\mathcal{L}_1$ training curve.

(b) $\mathcal{L}_2$ training curve.

(c) $\mathcal{L}_3$ training curve.

Figure 7: Training loss curves of RoCourseNet on the Loan dataset.

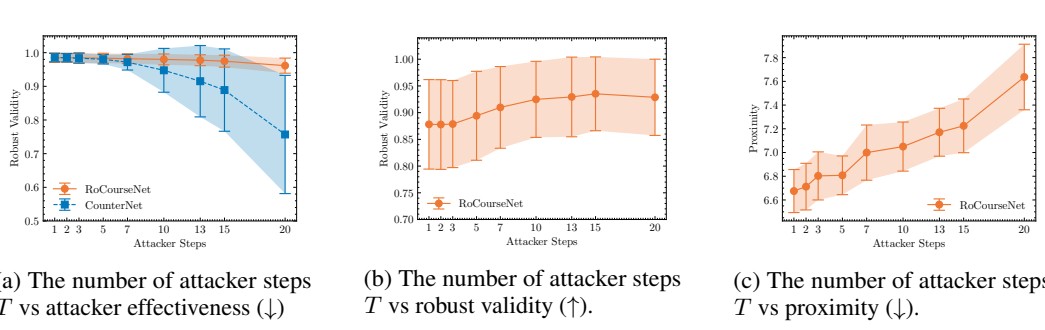

(a) The number of attacker steps $T$ vs attacker effectiveness ($\downarrow$)

(b) The number of attacker steps $T$ vs robust validity ($\uparrow$).

(c) The number of attacker steps $T$ vs proximity ($\downarrow$).

Figure 8: The impact of the number of attacker steps $T$ under the $l_2$-norm constrains.

**Experimental Results**    Table 7 shows that RoCourseNet achieves 100% validity and robust validity under both covariate and label shifts.

Table 7: RoCourseNet on simulated data shifts.

| Data Shift | Validity | Robust Validity | Proximity |
|------------|----------|-----------------|-----------|
| **Covariant** | 1.00 | 1.00 | $0.389 \pm 0.051$ |
| **Label** | 1.00 | 1.00 | $0.425 \pm 0.012$ |

## C    ABLATIONS OF ROCOURSENET

### C.1    TRAINING LOSS CURVE OF ROCOURSENET

Figure 7 shows RoCourseNet's training curve. Importantly, the prediction loss $\mathcal{L}_1$ and the proximity loss $\mathcal{L}_3$ are smoothly optimized during the training. The robust validity loss $\mathcal{L}_2$ encounters fluctuations in the early stage of training, but starts to converge after 10 epochs.

### C.2    $l_2$-NORM PROJECTION IN ALGORITHM 1

We provide supplementary results on adopting $\Delta$ as the $l_2$-norm ball (i.e., $\Delta = \{\delta \in \mathbb{R}^n \mid ||\delta||_2 \le \epsilon\}$) for the maximum perturbation constrains. Figure 8 highlights the results of using the $l_2$-norm ball in attacking and adversarial training. We observe similar patterns in Figure 3. Thus, this result shows that $l_\infty$-norm constrain can be substitute to other feasible region.

## D    TIME-COMPLEXITY ANALYSIS

### D.1    TRAINING TIME

RoCourseNet takes only $\sim$10 more minutes of training (as compared to CounterNet) on the Loan dataset (our largest-sized dataset). This is quite reasonable since training time is a one-time up-front

Table 8: Training time of CounterNet and RoCourseNet on the Loan dataset (the largest dataset).

| Dataset | CounterNet | RoCourseNet |
|---|---|---|
| Loan | 22m 6s | 32m 16s |
| German Credit | 46s | 1m 37s |
| Student | 55s | 2m 19s |

Table 9: Inference time for generating a single CF example on the Loan dataset (in milliseconds).

| Dataset | ROAR | RBR | CounterNet | RoCourseNet |
|---|---|---|---|---|
| Loan | 131.38 | 345.91 | 0.67 | 0.67 |
| German Credit | 104.87 | 271.34 | 0.51 | 0.51 |
| Student | 213.58 | 555.91 | 1.00 | 1.01 |

cost; after RoCourseNet is trained, test-time inference happens in milliseconds. Table 8 shows the training time of these two models.

## D.2 INFERENCE TIME

Inference runtime is an important metric as recourses are user-facing. Table 9 shows the inference runtime of RoCourseNet and baseline methods. Importantly, CounterNet and RoCourseNet achieve the same amount of inference time (as they share the same network structure). On the other hand, ROAR and RBR (two non-parametric methods) take significantly more time (i.e., $\sim$200X and $\sim$500X runtime as compared to RoCourseNet, respectively).

## E    DISCUSSION ABOUT GENERALIZED FRAMEWORKS TO ROCOURSENET

This section discusses extending the training of RoCourseNet (see Algorithm 1 & 2) into a broader framework. The training framework of RoCourseNet are not tied to the CounterNet architecture, and we can substitute predictor and CF generator components to other parametric models. Crucially, we can extend the RoCourseNet training into a general framework, which contains two components: (i) a predictor $f(\cdot; \theta)$, which makes an accurate prediction for a given input instance $x$, (ii) and a CF generator $g(\cdot; \theta_g)$, which generates its corresponding CF explanations. This general framework does not require a shared structure similar to the RoCourseNet architecture. We can choose separate models for the predictor $f(\cdot; \theta)$ and CF generator $g(\cdot; \theta_g)$. In addition, we can train this framework exactly as we train RoCourseNet via Algorithm 2, which optimizes for predictor $f(\cdot; \theta)$ and robust CF generator $g(\cdot; \theta_g)$.

**Experimental Settings.** We demonstrate the feasibility of training this general framework (denoted as *Robust CF Framework* in Table 10). For fair comparison, we use the same hyperparameters in training RoCourseNet (See Appendix A) to optimize this general framework. In addition, we use the same architecture specifications of RoCourseNet, as the predictor of this framework combines the encoder network and predictor network in RoCourseNet, and the CF generator network combines the encoder network and CF generator in RoCourseNet.

**Empirical Results.** Table 10 compares this general robust CF framework with RoCourseNet and Roar-LIME. This table highlights two important findings: (i) First, our proposed tri-level robust training (in Algorithm 2) can be extended to a general framework to optimize a robust CF generator. In particular, this robust CF framework outperforms Roar-LIME in terms of proximity and validity, and achieves the same level of robust validity as Roar-LIME. (ii) Additionally, we highlight the importance of RoCourseNet design (by leveraging the design of CounterNet (Guo et al., 2021)). This architecture design enables to generate well-aligned CF explanations by passing the predictor model's decision boundary (i.e., $p_x$) to the CF generator. From Table 10, we observe that RoCourseNet outperforms this Robust CF Framework in terms of the validity and robust validity, which underscores the importance of RoCourseNet's architecture designs.

Table 10: Evaluating robustness under model shift using a general framework.

| Methods | Metrics | | |
|---|---|---|---|
| | Proximity | Validity | Rob-Validity |
| **Robust CF Framework** | $7.150 \pm 1.078$ | $0.949 \pm 0.022$ | $0.906 \pm 0.119$ |
| **Roar-LIME** | $7.648 \pm 2.248$ | $0.937 \pm 0.046$ | $0.908 \pm 0.107$ |
| **RoCourseNet** | $7.183 \pm 0.406$ | $\mathbf{0.994 \pm 0.002}$ | $\mathbf{0.930 \pm 0.152}$ |

## F  DISCUSSION ABOUT MULTI-CLASS CLASSIFICATION

Existing CF explanation literature focuses on evaluating methods under the binary classification settings (Mothilal et al., 2020; Mahajan et al., 2019; Upadhyay et al., 2021; Guo et al., 2021). However, these CF explanation methods can be adapted to the multi-class classification settings. Given an input instance $x \in \mathbb{R}^d$, the RoCourseNet generates (i) a prediction $\hat{y}_x \in \mathbb{R}^k$ for input instance $x$, and (ii) a CF example $x^{\text{cf}}$ as an explanation for input instance $x$. The prediction $\hat{y}_x \in \mathbb{R}^k$ is encoded as one-hot format as $\hat{y}_x \in \{0,1\}^k$, where $\sum_i^k \hat{y}_x^{(i)} = 1$, $k$ denotes the number of classes. In addition, we assume a desired outcome $y'$ for every input instances $x$. As such, we can adapt Eq. 4 for binary settings to the multi-class settings as follows:

$$
\begin{aligned}
\operatorname*{argmin}_{\theta, \theta_g} \frac{1}{N} &\sum\nolimits_{(x_i, y_i) \in \mathcal{D}} \left[ \lambda_1 \cdot \underbrace{\mathcal{L}\Big(f(x_i; \theta), y_i\Big)}_{\text{Prediction Loss } (L_1)} + \lambda_3 \cdot \underbrace{\mathcal{L}\Big(x_i, x_i^{\text{cf}}\Big)}_{\text{Proximity Loss } (L_3)} \right] \\
&+ \max_{\boldsymbol{\delta}, \forall \delta_i \in \Delta} \frac{1}{N} \sum\nolimits_{(x_i, y_i) \in \mathcal{D}} \left[ \lambda_2 \cdot \underbrace{\mathcal{L}\Big(f\left(x_i^{\text{cf}}; \theta'_{opt}(\boldsymbol{\delta})\right), y'\Big)}_{\text{Robust Validity Loss } (L_2)} \right] \\
s.t \ \ \theta'_{opt}(\boldsymbol{\delta}) &= \operatorname{argmin}_{\theta'} \frac{1}{N} \sum\nolimits_{(x_i, y_i) \in \mathcal{D}} \left[ \mathcal{L}\Big(f(x_i + \delta_i; \theta'), y_i\Big) \right], \ x_i^{\text{cf}} = g(x_i; \theta_g).
\end{aligned}
\tag{5}
$$

To optimize for Eq. 5, we can follow the same procedure outlined in Algorithm 2. For each sampled batch, we first optimize for the predictive accuracy $\theta' = \theta - \nabla_\theta(\lambda_1 \cdot L_1)$. Next, we use the VDS algorithm to optimize for the inner max-min bi-level problem (in Algorithm 1). Finally, we optimize for the CF explanations by updating the model's weight as $\theta''_g = \theta''_g - \nabla_{\theta'_g}(\lambda_2 \cdot L_2 + \lambda_3 \cdot L_3)$.

