# OpenReview forum: "RoCourseNet: Distributionally Robust Training of a Prediction Aware Recourse Model"
_ICLR.cc/2023/Conference — Submitted to ICLR 2023_

### Official Review · Reviewer_KsVC · 2022-10-16

**Confidence:** 4
**Correctness:** 3
**Technical Novelty And Significance:** 2
**Empirical Novelty And Significance:** 2
**Recommendation:** 3

**Clarity, Quality, Novelty And Reproducibility:**

The quality of this submission is not high. The motivation is not very clear. The evaluation metrics are not comprehensive.

Some notations are not clear.

The proposed methods seem original.

**Strength And Weaknesses:**

Strength:

1. The paper is easy to follow.

Weakness:

1. The motivation is not clear. When I look at the title of this submission, I feel this paper is related to the distributionally robust optimization (DRO) topic. However, I find the main part of this paper is related to adversarial perturbations. Therefore, the title is easy to mislead the understanding of this paper. I do not mean this title is a mismatch with this paper. But 'DISTRIBUTIONALLY ROBUST' is not fit well in this paper. Most importantly, it seems like the robust recourse is dynamically generated during training. According to (4), $x^{cf}$ is generated by another model (i.e., neural network) $g$. The final model should be $f$. I guess the distribution shift only works on $x$ instead of $x^{cf}$. However, in Figure 1, it is not clear what the data shift is. One observation is the model shifted but the data is not shifted. Moreover, the experiments are based on three datasets. However, distributional shifts in the training data (as mentioned in the Abstract) are not demonstrated on these datasets.

2. Notations are not clear. For example, in Section 3.1, what is the domain of $f$? Is it a real number or belongs to {0,1}? In addition, below (1), $\mathcal{F}={\theta'|\theta+\delta}$. This $\delta$ works on the model parameters. However, according to (3), $\delta$ seems working on features. I am very confused with this parameter.

3. In experiments, Table 1 is hard to read. For the three metrics, it is not clear whether the values large better or small are better. Most importantly, what is the meaning of bold values? If the bold value means the best result, why 6.746 $\pm$ 0.723(from Loan, Proximity) is the best? It seems like this value is not the best.

4. In Section 4 (Cost-Validity Trade-Off), I am very confused with the discussion from the **Cost-Validity Trade-Off** paragraph. First, I am not sure how to define 'the three best performing models from Table 1'. Section, What is the meaning of ambiguity sizes $\epsilon_1$ $\epsilon_2$? Third, the authors mentioned 'These figures show that ...'. I do not know which figure is for this discussion.

5. Since the proposed optimization framework has no convergency guarantee, the authors should show the tendency curve of training loss. Moreover, since this task is a classification task, the authors should show the accuracy performance besides the metrics only for CF. Even Upadhyay et al., 2021 show accuracy and AUC (See Table 2 in [Upadhyay et al., 2021] for evaluating the performance of generating robust recourse). Without accuracy, it is hard to say the applicability of the proposed model.

**Summary Of The Paper:**

This submission proposed an approach to solving the data distributional shift problem in recourse models. A min-max-min learning objective is proposed to learn the predictions and enhance the robustness of future data shifts. Then an optimization framework, RoCourseNet, is designed to optimize the learning objective. Experiments on three datasets and comparisons with several existing baselines demonstrate the effectiveness of the proposed method.

**Summary Of The Review:**

I suggest rejecting this paper. For details, please see my weakness.

---

> ### Author Response · Authors · 2022-11-17
> **Response to Reviewer KsVC**
>
> We thank the reviewer for their insightful and detailed feedback. Wherever possible, we incorporate your suggestions into our paper. Please read answers to individual questions below.
>
> > On the title
> >
>
> Thank you for that suggestion. To avoid confusion, we propose to change the paper title to “RoCourseNet: Robust Training of a Prediction Aware Recourse Model”. This purposed title is reflected in the new paper draft.
>
> > On Figure 1
> >
>
> We apologize for the confusion caused in Figure 1. We have updated the new Figure 1 for improved clarity. As seen in this new figure, (i) given an ML model (represented by its decision boundary $f(.;\theta)$) CF generation methods try to find a valid CF for point x that lies on the opposite side of $f(.;\theta)$. (ii) when new training data becomes available, this causes the overall distribution of training data to shift, which causes a shift in the ML model (represented by an updated decision boundary $f(.;\theta’)$.
>
> Similarly, in each of our evaluation datasets, “distributional shifts” occur because we get new training data over time. Every time we get new training data, that leads to a distributional shift in the training data distribution (by definition).
>
> > On Notation
> >
>
> In Section 3.1, The domain of $f$ is $R^d$ where $d$ is the number of features in our dataset. As for delta, thank you for pointing out that we have overloaded the use of delta. We have updated our paper as $\mathcal{F} = \{\theta' \ | \ \theta + \delta_f \}$ for improved notation.
>
> > On Experiments
> >
>
> Thank you for pointing out the typo. We have updated the paper based on your suggestions.
>
> For further clarification, in Table 1, a **lower** proximity score is more desirable, but **higher** validity and **higher** robust validity are better. For each dataset, bold values highlight the best-performing model in terms of a particular metric. For example, on the Loan dataset, RoCourseNet achieved the best (**lowest**) proximity score of $6.611\pm0.418$ (hence, we have put it in bold).
>
> > On Cost-Validity Tradeoff
> >
>
> Thank you for your careful reading of our paper. To address your concerns, we revised the sentence “the three best performing models from Table 1” into “three recourse methods explicitly optimizing for distributional shift”. Secondly, the ambiguity sizes are hyperparameters in the RBR model (which we specify in the updated paper). Finally, we revised the sentence “These figures show that…” into “Figure 2 shows that …”.
>
> We have updated our presentation of the Cost Invalidity trade-off based on your feedback.
>
> > On Training Loss & Accuracy
> >
>
> Thank you for that suggestion. We have added a figure on training Loss in Appendix C (see Figure 7). In addition, we have also added numbers on accuracy and AUC for RoCourseNet and baseline approaches in Appendix B (see Table 4 & 5).

---

### Official Review · Reviewer_N727 · 2022-10-24

**Confidence:** 3
**Correctness:** 4
**Technical Novelty And Significance:** 2
**Empirical Novelty And Significance:** 2
**Recommendation:** 3

**Clarity, Quality, Novelty And Reproducibility:**

- This paper is quite clearly written, and the technical quality is high. However the novelty aspect of the paper is relatively low.


**Strength And Weaknesses:**

Strengths:
- The method is fairly straightforward and easy to understand, combining adversarial training with recourse generation.

- The method works better than other state-of-art method in robust recourse generation, and the experimental evaluation is quite thorough.

Weaknesses:
- The method is a fairly direct combination of robustness through adversarial training and existing recourse generation methods. The novelty is not very high, although the authors did solve the technical difficulty of solving the tri-level optimization problem.

- The current work considers robustness to adversarial perturbations to inputs in generating counterfactual explanations. But adversarial perturbations is a fairly specific and strong type of data shift, which does not cover all the cases suggested in the introduction. Have the authors considered robustness to other types of common data shifts such as covariate shifts or label shifts?

- It is odd that while the algorithm is trained on adversarial perturbations, the evaluation is focused on temporal data shifts, which is a mismatch. Are there any explanations on why models trained on adversarial data shifts work well on temporal data shifts?


**Summary Of The Paper:**

The authors propose combining adversarial training with recourse generation methods to produce a robust recourse generation method, by solving a min-max-min tri-level optimization problem. They show that their method works better than competing state-of-art methods on several real world data shift datasets.


**Summary Of The Review:**

This work is of high technical quality and the method produced by the authors can beat other state-of-art methods in generating robust recourse. The main limitation is the relative lack of novelty of the method or any new understanding or insights.

---

> ### Author Response · Authors · 2022-11-17
> **Response to Reviewer N727**
>
> We thank the reviewer for their insightful and detailed feedback. Below, we provide clarifying answers to all your questions.
>
> > Clarifying the novelty of the paper
> >
>
> We apologize for not making the novelty of our paper clearer. You are right in pointing out that the ideas of adversarial training and recourse generation methods are not new. However, it is highly non-trivial to combine these ideas in a way that leads to highly robust recourses, and that has never been attempted before, to the best of our knowledge.
>
> The main technical novelty of our work lies in the formulation of a novel tri-level optimization problem for robust recourse generation. Unlike previous work, our formulation explicitly accounts for shifts in the underlying training data distribution, and connects them to corresponding shifts in the ML model parameters. **To the best of our knowledge, ours is the first tri-level optimization formulation for generating robust recourses**. As you mentioned in your review, solving this tri-level optimization problem is very challenging.
>
> To solve this tri-level problem, we proposed a highly non-trivial adversarial training scheme (Algorithm 2) that is designed specifically for CounterNet’s coordinate descent approach to training. Further, to model our attacker, we proposed a Virtual Data Shift algorithm which relied on an unrolling optimization pipeline that finds the worst-case data shift that leads to an adversarially shifted model.
>
> Finally, as mentioned in the paper, all existing approaches for robust recourse generation are post-hoc in nature (which represents the dominant paradigm for generating robust CF explanations). In this work, we make a departure from this prevalent post-hoc paradigm, as we make the first-ever novel attempt at developing an end-to-end robust recourse generation method in which the robust recourses are jointly trained with the prediction model. As shown empirically, this departure from the post-hoc paradigm leads to SOTA results.
>
> Also, our rigorous experimental evaluation of a wide variety of baseline methods shows that RoCourseNet achieves SOTA performance on several popular evaluation metrics.
>
> > Algorithm trained on adversarial perturbations, but evaluated on temporal data shifts
> >
>
> There seems to be some confusion.
>
> First, our evaluation section is not only limited to temporal data shifts, we also consider spatial shifts and data correction shifts (as mentioned in Section 4).
>
> Second, the algorithm is trained for adversarial perturbations of any kind, so in theory, RoCourseNet can handle different kinds of data shifts. By saying that RoCourseNet is trained on adversarial perturbations, we mean to say that RoCourseNet does not know a priori what kind of perturbation is going to occur in the real-world dataset, and thus, the algorithm makes a worst-case assumption and tries to optimize against the worst-case perturbations.
>
> Since the dataset that we evaluate has temporal data shifts, however, we do not know what specific shift occurs in our temporal dataset. Therefore, RoCourseNet assumes and optimizes against **worst-case (or adversarial) temporal data shifts**. Thus, there is no mismatch.
>
> > On covariate shift and label shift
> >
>
> Thank you for that suggestion. As mentioned above, RoCourseNet can easily handle any kind of shift by making the adversarial perturbation assumption. So, we can easily handle covariate shifts and label shifts, etc.
>
> To further address your concern, we have added simulated experiments with covariate shift and label shift into the Appendix of the paper. Specifically, we create simulated datasets with covariate and label shifts and show how well RoCourseNet performs on them. As shown below, RoCourseNet achieves 100% validity and robust validity under both covariate and label shifts. We elaborate on the simulated data shift in Appendix B3.
>
> |  | Validity | Robust Validity | Proximity |
> | --- | --- | --- | --- |
> | Covariant Shift | 1.00  | 1.00  | 0.389 $\pm$ 0.051 |
> | Label Shift | 1.00  | 1.00  | 0.425 $\pm$ 0.012 |

---

### Official Review · Reviewer_GDb3 · 2022-10-24

**Confidence:** 4
**Correctness:** 3
**Technical Novelty And Significance:** 2
**Empirical Novelty And Significance:** Not applicable
**Recommendation:** 3

**Clarity, Quality, Novelty And Reproducibility:**

The paper explanation of the method is clear with the nice addition of suitable pseudo algorithms.
Code is provided as well, which helps reproducibility.

However caption 1 is insufficient and should be more detailed.
The same can be said basically for all captions in this paper.

In table 1, first row, first column, is Counternet the right value to be put in bold?

Regarding novelty, while the authors are the first to consider this problem, it is not clear if the problem itself is rightly formulated or interesting enough in this form, limiting the overall impact.


**Strength And Weaknesses:**

## Strenghts:

The authors are effective in inserting their work in the current state of the art, with a satisfactory related work and introduction section.
The methodology introduced is clear with helpful pseudo algorithms as accompaniment.
The authors include anonymized code with their submission, which helps in the reproducibility compartment.


## Weaknesses:

My main concern with this paper is the intuitive explanation of the underlying problem and, as a result, its lack of a formal definition.

The authors provide the classic example of an applicant to a loan bank.
Their goal is to provide a counterfactual explanation to a rejected applicant that is “robust” in the sense that the following time, if the conditions are met, the applicant will see its loan accepted.
What is then the definition of “robust” in this case? What if interest rates drastically change in the meanwhile? Should the application still be accepted? Or is it more a sort of “ceteris paribus” change considered here?

This leads to the next point. The authors consider a worst case scenario, and use an adversarial technique to represent this.
It is the opinion of this reviewer that the authors did not really justify this choice.
As seen in the literature (Goodfellow 2014 and many others), adversarial attacks are often random to the human eye. Why then assume a worst case scenario, maybe even an unfeasible one? What is the probability of observing such a change?

Finally, it could be argued that this method  ends up just trading off validity and robust validity with an increased proximity, as it seems to be the case in at least two of the three datasets considered in figure 2.

The fact that all comparisons are performed against methods that make stronger assumptions does not help in evaluating the method, the addition of a heuristic based similar technique would have helped in this regard (even a simple random baseline).


**Summary Of The Paper:**

The paper tackles the problem of generating counterfactual examples that are robust to distribution shifts.
To this regard, the authors first formulate a framework for this problem that considers a tri-level optimization problem.
Second, they propose a methodology, RoCourseNet, that solves a tri-level optimization problem.
They perform experiments on three standard benchmarks and find their results to be highly robust counterfactual explanations against data shifts that consistently outperform previous state of the art.


**Summary Of The Review:**

While the paper has some strengths, as highlighted in the relevant strength section, the lack of a clear and formal definition of the problem they are facing severely limits their chance of acceptance for this paper as it is.
Addressing the points listed in the weaknesses section could improve the rating of this paper according to this reviewer.

---

> ### Author Response · Authors · 2022-11-17
> **Response to Reviewer GDb3 (Part 2)**
>
> > Why assume a worst-case scenario for adversarial training?
> >
>
> Assuming worst-case scenarios is a very common assumption for situations when there is no prior information/belief about the specific kind of attacks (or perturbations) that we expect to observe in the real-world [3]. In such situations, assuming the worst case enables you to prepare for any kind of attack or perturbation (e.g., worst-case adversarial attack, random perturbation attack, etc.) that may happen in the future. Thus, our worst-case assumption enables us to be prepared not only against attacks that may appear random to the human eye, but also any other possible attacks.
>
> For datasets in the wild, we don’t know a priori what kinds of data (or model) shifts we can expect. More importantly, each dataset may experience a different kind of shift. Since we want to build a general-purpose robust CF explanation system that can generate robust CF explanations for any dataset, the only reasonable choice for us was to assume and plan against a worst-case scenario, so that our robust CF explanations would be robust against any possible data shift (and corresponding model shift) that may occur.
>
> To further address your concern, we compare RoCourseNet against RoCourseNet-Random, a variant of CounterNet which is optimized against a random perturbation attacker. Below, we show results obtained on the Loan application dataset, which shows that RoCourseNet achieves ~14% higher validity, ~24% higher robust validity, and ~58% better proximity as compared to RoCourseNet-Random. This shows that perturbations on our three real-world datasets do not appear to be random perturbations, since our worst-case assumption enables us to achieve significantly better metrics.
>
> |  | Validity | Robust Validity | Proximity |
> | --- | --- | --- | --- |
> | VanillaCF-Random | 0.6343$\pm$0.270 | 0.5101$\pm$0.251 | 9.446$\pm$1.042 |
> | RoCourseNet-Random | 0.856 $\pm$ 0.127 | 0.713 $\pm$ 0.265 | 10.405 $\pm$ 1.857 |
> | RoCourseNet | 0.996 $\pm$ 0.002 | 0.969 $\pm$ 0.106 | 6.611 $\pm$ 0.418 |
>
> We have added these experiments to the Appendix B.2 (Table 6).
>
> > RoCourseNet trades off validity and robust validity with an increased proximity
> >
>
> We share this intuition. In fact, Rawal et al. [4] theoretically demonstrate this cost-invalidity trade-off. RoCourseNet optimally trades off validity and robust validity with proximity to achieve good performance across all these metrics.
>
> > Addition of heuristic baselines
> >
>
> In addition to RoCourseNet-Random, we have also implemented VanillaCF-Random (another heuristic baseline in which we add a random perturbation to the recourse generated by VanillaCF. The comparison results of these two baselines against RoCourseNet can be found in Appendix B.2 (Table 6).
>
> ## References
> [3] Madry, A., Makelov, A., Schmidt, L., Tsipras, D., & Vladu, A. (2019). Towards Deep Learning Models Resistant to Adversarial Attacks. *ArXiv:1706.06083 [Cs, Stat]*. [http://arxiv.org/abs/1706.06083](http://arxiv.org/abs/1706.06083)
>
> [4] Rawal, K., Kamar, E., & Lakkaraju, H. (2020). Algorithmic recourse in the wild: Understanding the impact of data and model shifts. *arXiv preprint arXiv:2012.11788*

---

> ### Author Response · Authors · 2022-11-17
> **Response to Reviewer GDb3 (Part 1)**
>
> We thank the reviewer for their insightful and detailed feedback. We have fixed all captions and typos in the paper as suggested by you. Below, we provide clarifying answers to all your questions.
>
> > On the definition of robustness
> >
>
> Our definition of robustness is not new. Both our baselines (ROAR [1] and RBR [2]) use the exact same definition of robustness. We clarify this definition below.
>
> Let input data point $x$ denote a loan application that gets denied by the bank’s ML algorithm. Then, a robust CF example $x^\text{cf}$ (for input x) is one that remains valid (i.e., recourse loan application $x^\text{cf}$ gets approved by the bank’s ML algorithm) even if this bank’s ML algorithm changes over time.
>
> So let the original ML model be a parametric model $f(.;\theta)$. If we generate a robust CF example $x^\text{cf}$, then (i) we want it to be valid on any new updated (or shifted) ML model. So it may be the case that in two years from now, the bank updates its ML model from $f(.;\theta)$ to $f(.;\theta’)$. And if we generate a robust CF example $x^\text{cf}$, we want $x^\text{cf}$ to be valid on any possible $f(.;\theta’)$ that we may see in the future.
>
> Note that this change in the bank’s ML model may be caused due to a variety of real-world factors, including things like changes in interest rates (as you mentioned), changes in the bank’s outlook on market conditions, changes in management, etc.
>
> We reiterate the formal definition of robustness from Section 3.1. For an input $x$ and an original ML model $f(.;\theta)$, a generated CF example $x^\text{cf}$ is robustly valid w.r.t. a shifted model $f(.; θ′)$ iff $x^\text{cf}$ gets an opposite prediction from the shifted model (as compared to the prediction received by x on the original model $f(.;\theta)$. In short, we want that $f (x^\text{cf}; \theta') = 1 − f (x; \theta)$.
>
> We hope that this clarifies any confusion on the definitions of robustness used in the paper.
>
> ## References
>
> [1] Upadhyay, S., Joshi, S., & Lakkaraju, H. (2021, May 21). *Towards Robust and Reliable Algorithmic Recourse*. Thirty-Fifth Conference on Neural Information Processing Systems.
>
> [2] Nguyen, T. D. H., Bui, N., Nguyen, D., Yue, M. C., & Nguyen, V. A. (2022, August). Robust Bayesian Recourse. In *Uncertainty in Artificial Intelligence* (pp. 1498-1508). PMLR.

---

### Official Review · Reviewer_hKo5 · 2022-10-29

**Confidence:** 4
**Correctness:** 4
**Technical Novelty And Significance:** 2
**Empirical Novelty And Significance:** 2
**Recommendation:** 6

**Clarity, Quality, Novelty And Reproducibility:**

The paper is written clearly and is easy to follow. Code and implementation details are provided for reproducibility. The paper builds on existing work (CounterNet [1]), and novelty is in learning the ‘adversarial model’ for which they propose the VDS algorithm.

**Strength And Weaknesses:**

+ The paper deals with a practical problem; generating robust recourses is necessary for models which are to be deployed in the real world
+ RoCourseNet outperforms the baselines convincingly in generating robust recourses for the 3 datasets considered
+ RoCourseNet works with the full model and not its locally linear approximation (via LIME etc.) which allows it to model larger number of model shifts via the VDS algorithm

- RoCourseNet involves a tri-level optimization problem. How much additional computational effort does ReCourseNet require? A comparison of the training time taken vs CounterNet seems necessary.
- The method lacks some flexibility of post-hoc counterfactual generation methods. Ex, different people have different notions of cost (proximity) or actionability. Can RoCourseNet solve this without retraining?

Other points:
* In Algorithm1 VDS line (8) how is this gradient w.r.t \delta computed? Is the only dependence of \delta through \theta(\delta)?
* Although not completely fair, a comparison of the training time w.r.t ROAR [2] may also be instructive.
* Cite the published version of ROAR

[1] Hangzhi Guo, Thanh Nguyen, and Amulya Yadav. Counternet: End-to-end training of counterfactual aware predictions. In ICML 2021 Workshop on Algorithmic Recourse, 2021.
[2] Upadhyay, Sohini, Shalmali Joshi and Himabindu Lakkaraju. “Towards Robust and Reliable Algorithmic Recourse.” NeurIPS (2021).

**Summary Of The Paper:**

The paper proposes an architecture and a training methodology (termed RoCourseNet) to generate a robust counterfactual (cf) along with the prediction for a given factual point. RoCourseNet builds on earlier work CounterNet [1] by modifying its objective to generate robust cfs i.e cfs which stay valid even when the underlying model shifts. As part of the RoCourseNet objective, the inner ‘adversary’ itself is proposed as a bilevel problem (called VDS in the paper). The paper proposes to learn a ‘worst-case’ classifier by looking at how the training dataset can change such that a classifier learnt on this new dataset maximally invalidates the old cfs.  Experiments are performed on 3 real-world datasets and they compare against 4 baselines.


**Summary Of The Review:**

The paper solves an important problem. The experimental protocol and results are convincing. The main issue I have is I feel the method is computationally expensive, and it lacks some flexibility that post-hoc cf-generation methods have. I propose acceptance, conditional on some time-complexity analysis.

---

> ### Author Response · Authors · 2022-11-17
> **Response to Reviewer hKo5**
>
> Thank you so much for your insightful comments and feedback. We have updated our paper to incorporate all of your suggestions. Below, we provide answers to your questions.
>
> > Training time analysis
> >
>
> RoCourseNet takes only ~10 more minutes of training (as compared to CounterNet) on the Loan dataset (our largest-sized dataset). This is quite reasonable since training time is a one-time up-front cost; after RoCourseNet is trained, test-time inference happens in milliseconds. Nevertheless, as suggested by you, we have added the training time comparison of CounterNet and RoCourseNet in the Appendix (in Table 8).
>
> Here, we show the training time of CounterNet and RoCourseNet on the largest-sized dataset (i.e., Loan dataset). Please refer to Table 8 in the Appendix for training time in other two datasets.
>
> |  | CounterNet | RoCourseNet |
> | --- | --- | --- |
> | Training Time | 22m 6s | 32m 16s |
>
> > Runtime comparison with baselines
> >
>
> Both ROAR and RBR are not parametric model-based methods, hence, there is no analog of training time for ROAR and RBR. However, we do compare the inference time (i.e., time taken to generate a CF example for a single data point) of RoCourseNet, CounterNet, ROAR and RBR (see Table 9 in Appendix D).
>
> Here, we show the inference time on the largest-sized dataset (i.e., Loan dataset). Please refer to Table 9 in Appendix D for the inference time in other two datasets.
>
> |  | CounterNet | RoCourseNet | ROAR | RBR |
> | --- | --- | --- | --- | --- |
> | Test Inference Time | 0.67 ms | 0.67 ms | 131.38ms | 345.91ms |
>
> > Lacked flexibility than post-hoc methods.
> >
>
> We do agree that in order to account for different user-specific notions of cost/proximity, RoCourseNet currently needs to be retrained. However, note that this lack of flexibility is not unique to RoCourseNet, e.g., the same limitation would apply to any model-based CF generation method. We consider extending RoCourseNet to handle user-specific notions of cost as future work.
>
> As a first step, we tried to implement a naive RoCourseNet variant for handling user-specific notions of cost, which would not require retraining RoCourseNet from scratch. Specifically, we tried the following two-step approach:
>
> 1. We train RoCourseNet normally as described in the paper.
> 2. For any new data point (user) which has a different notion of cost, we can fine-tune RoCourseNet with the user-specific notion of cost for a single epoch.
>
> We compared this fine-tuning variant of RoCourseNet (i.e., *RoCourseNet-Finetune*) against another variant in which we fully retrain the model (i.e., *RoCourseNet-Weighted*) from scratch, and the original RoCourseNet (i.e., *RoCourseNet-Unweighted*). Our result shows that one-epoch fine-tuning is not sufficient to achieve desirable weighted proximity. This negative result demonstrates that a non-trivial adaptation is required for RoCourseNet to handle user-specific cost/proximity notions.
>
> |  | Robust Validity | Weighted Proximity |
> | --- | --- | --- |
> | RoCourseNet-Unweighted | 0.965 $\pm$ 0.113 | 9.801$\pm$2.918 |
> | RoCourseNet-Finetune | 0.957$\pm$0.183 | 9.754$\pm$2.084 |
> | RoCourseNet-Weighted | 0.918$\pm$0.194 | 6.823 $\pm$0.994 |
>
> > In Algorithm1 VDS line (8) how is this gradient w.r.t $\delta$ computed? Is the only dependence of $\delta$ through $\theta(\delta)$?
> >
>
> Yes, the gradient w.r.t. $\delta$ depends on $\theta(\delta)$, where $\theta(\delta)$ is a function resulting from LINE 5-7 in Algorithm 1. We have updated our paper with elaborations on this point.
>
> Finally, we have also updated the ROAR reference.

---

### Author Response · Authors · 2022-11-18
**Rebuttal Summary**

We appreciate the time and feedback from reviewers. We have significantly updated our paper based on suggestions by the reviewers, and have added several additional experiments to the updated paper and Appendix (as suggested reviewers). Reviewers can view the revision history on this [page](https://openreview.net/revisions?id=zufPou5foW) or check the exact changes via this [link](https://openreview.net/revisions/compare?id=zufPou5foW&left=MVrY7XJMLX&right=BO1uGKC3i6&pdf=true).

Here, we provide a brief summary of the key points raised by reviewers. Rebuttals to individual reviewers can be found after each review.

## Summary

1. **Reviewer [hKo5](https://openreview.net/forum?id=zufPou5foW&noteId=C6xz9G0pRf)** wrote “**I** ***propose acceptance**, conditional on the addition of time-complexity analysis*” (we have duly added these new results in the appendix, see our detailed [response](https://openreview.net/forum?id=zufPou5foW&noteId=t4aqIoyKRYU) to **Reviewer [hKo5](https://openreview.net/forum?id=zufPou5foW&noteId=C6xz9G0pRf)** for further details).
2. **Reviewer [N727](https://openreview.net/forum?id=zufPou5foW&noteId=2LsyxIaEBNi)** was under the impression that “*this work is of **high technical quality** and the **method produced by the authors can beat other state-of-art methods in generating robust recourse***.” They had confusions about the novelty of the paper’s contributions and our usage of the term “adversarial perturbations”, but we have written a detailed [response](https://openreview.net/forum?id=zufPou5foW&noteId=OF2UQE34YH9) to address their concerns.
3. **Reviewer [GDb3](https://openreview.net/forum?id=zufPou5foW&noteId=m-GXnLW_A_)** felt that “***our paper had strengths**, and that we were **effective in inserting our paper in the current state-of-the-art***”. They also said that “*they would be **willing to increase score** if their questions are answered”,* which mainly revolved around getting a clear definition of robustness, getting a justification for worst-case assumptions used in the paper, and seeing some additional heuristic baselines implemented in the paper. In response, we have updated our description of robustness in the paper, we provide a detailed response to justify our worst-case assumption, and we also implement two heuristic baselines as suggested by them (see our detailed [response](https://openreview.net/forum?id=zufPou5foW&noteId=Zee1GHV2kD) to **Reviewer [GDb3](https://openreview.net/forum?id=zufPou5foW&noteId=m-GXnLW_A_)**).
4. **Reviewer [KsVC](https://openreview.net/forum?id=zufPou5foW&noteId=zoGhFXCYWM)** felt that “***our proposed methods seem original***”, but they correctly pointed out some typos in notations, in our tables, and in our overall writing. We fixed all typos and notations as suggested by them, and improved the writing to make it clearer. In addition, they suggested a change in the paper title, which we have done. Finally, they asked for the addition of prediction comparisons in the paper, and we have duly added these additional experiments in the Appendix (see the detailed response to **Reviewer [KsVC](https://openreview.net/forum?id=zufPou5foW&noteId=gomwNYtwsOI)** below).

---

### Decision · Program_Chairs · 2023-01-20

**Decision:**

Reject

**Justification For Why Not Higher Score:**

Basically nobody was very excited about this paper. One reviewer found the problem very interesting, but nobody loved the solution or found the results overwhelming.

**Justification For Why Not Lower Score:**

N/A

**Metareview: Summary, Strengths And Weaknesses:**

This paper provides a method for explaining machine learning classifiers using counterfactuals. In particular, they are especially interested in being robust to distribution shifts.

The reviewers had mixed feelings on many aspects of the paper: the importance of the problem, the motivation for using an adversarial approach, and the practicality of the given approach.